# Construction and Validation of a Reliable Disulfidptosis-Related LncRNAs Signature of the Subtype, Prognostic, and Immune Landscape in Colon Cancer

**DOI:** 10.3390/ijms241612915

**Published:** 2023-08-18

**Authors:** Xiaoqian Dong, Pan Liao, Xiaotong Liu, Zhenni Yang, Yali Wang, Weilong Zhong, Bangmao Wang

**Affiliations:** 1School of Medicine, Nankai University, Tianjin 300071, China; 1120220806@mail.nankai.edu.cn (X.D.);; 2Department of Gastroenterology and Hepatology, Tianjin Medical University General Hospital, Tianjin Institute of Digestive Diseases, Tianjin Key Laboratory of Digestive Diseases, Tianjin 300052, China

**Keywords:** disulfidptosis, long noncoding RNA (lncRNA), colon cancer, molecular subtype, prognostic signature, immune microenvironment, drug sensitivity

## Abstract

Disulfidptosis, a novel form of regulated cell death (RCD) associated with metabolism, represents a promising intervention target in cancer therapy. While abnormal lncRNA expression is associated with colon cancer development, the prognostic potential and biological characteristics of disulfidptosis-related lncRNAs (DRLs) remain unclear. Consequently, the research aimed to discover a novel indication of DRLs with significant prognostic implications, and to investigate their possible molecular role in the advancement of colon cancer. Here, we acquired RNA-seq data, pertinent clinical data, and genomic mutations of colon adenocarcinoma (COAD) from the TCGA database, and then DRLs were determined through Pearson correlation analysis. A total of 434 COAD patients were divided in to three subgroups through clustering analysis based on DRLs. By utilizing univariate Cox regression, the least absolute shrinkage and selection operator (LASSO) algorithm, and multivariate Cox regression analysis, we ultimately created a prognostic model consisting of four DRLs (AC007728.3, AP003555.1, ATP2B1.AS1, and NSMCE1.DT), and an external database was used to validate the prognostic features of the risk model. According to the Kaplan–Meier curve analysis, patients in the low-risk group exhibited a considerably superior survival time in comparison to those in the high-risk group. Enrichment analysis revealed a significant association between metabolic processes and the genes that were differentially expressed in the high- and low-risk groups. Additionally, significant differences in the tumor immune microenvironment landscape were observed, specifically pertaining to immune cells, function, and checkpoints. High-risk patients exhibited a low likelihood of immune evasion, as indicated by the Tumor Immune Dysfunction and Exclusion (TIDE) analysis. Patients who exhibit both a high risk and high Tumor Mutational Burden (TMB) experience the least amount of time for survival, whereas those belonging to the low-risk and low-TMB category demonstrate the most favorable prognosis. In addition, the risk groups determined by the 4-DRLs signature displayed distinct drug sensitivities. Finally, we confirmed the levels of expression for four DRLs through rt-qPCR in both tissue samples from colon cancer patients and cell lines. Taken together, the first 4-DRLs-based signature we proposed may serve for a hopeful instrument for forecasting the prognosis, immune landscape, and therapeutic responses in colon cancer patients, thereby facilitating optimal clinical decision-making.

## 1. Introduction

Colon cancer is a significant global health concern, representing a substantial burden on both patients and healthcare systems. According to the American Society of Clinical Oncology (ASCO), approximately 104,270 people in the United States are expected to receive a colon cancer diagnosis in 2021 [1]. Early detection and accurate prognostication play pivotal roles in improving patient outcomes. Traditional clinical factors, such as tumor stage, nodal involvement, and distant metastasis, are extensively employed to forecast prognosis and assist in making treatment choices [2]. However, advancements in molecular biology and high-throughput technologies have unveiled additional layers of complexity in CRC, highlighting the need for more refined predictive models. Hence, it is crucial to develop dependable and efficient predictive biomarkers in order to recognize unique subsets among individuals with colon cancer. These biomarkers would serve as indispensable tools in guiding personalized and optimal treatment strategies. Given the rapid advancements in bioinformatics, it becomes even more critical to leverage these technologies for the identification and validation of such biomarkers.

Disulfidptosis, a newly characterized type of regulated cell death (RCD) associated with metabolic alterations, plays a multifaceted function in the realm of anti-tumor immune response [3]. Importantly, a recent investigation revealed that increased SLC7A11 expression can impede ferroptosis when there is a lack of glucose by aiding in the absorption of cystine. However, this process may trigger the occurrence of disulfidptosis [4,5]. Distinct from apoptosis and ferroptosis, disulfidptosis is not influenced by other cell death inhibitors. Instead, it is specifically intensified by thiol oxidation reagents like diamide [5]. Consequently, the emergence of disulfidptosis may holds potential for novel avenues in tumor treatment. Furthermore, studies have demonstrated that disulfidptosis also possesses the capacity to impact immune infiltration [6]. However, there is a need for the identification and establishment of more biomarkers related to disulfidptosis, and a comprehensive understanding of its underlying mechanisms and therapeutic implications necessitates further research and exploration.

As key participants in cancer biology, long non-coding RNAs (lncRNAs) provided new understandings into tumorigenesis and possible therapeutic paths. Unlike protein-coding genes, lncRNAs exert their impact through various mechanisms, such as chromatin alteration, regulation of transcription, post-transcriptional modification, and protein interaction [7]. Aberrant expression of specific lncRNAs has been linked to various aspects of colorectal carcinogenesis, including cell proliferation, apoptosis resistance, epithelial-mesenchymal transition (EMT), and immune evasion [8,9]. Moreover, dysregulation of lncRNAs has also been linked to various clinical features, such as tumor stage, lymph node involvement, occurrence of metastasis, and patient prognosis [10]. Expanding the knowledge of lncRNA’s involvement in colon cancer has paved the way for their potential clinical applications. The potential of these lncRNAs as biomarkers for early detection, risk stratification indicators, and treatment response predictors in patients with colon cancer is highly promising [11]. Additionally, targeting specific dysregulated lncRNAs may offer a novel therapeutic strategy for combating colon cancer and overcoming drug resistance [12]. However, there is still a lack of studies investigating the relationship between disulfidptosis and lncRNA in colon cancer.

This research established a dependable disulfidptosis-related lncRNAs (DRLs) signature for forecasting prognosis and guide clinical treatment. Initially, a predictive pattern was developed using 4 DRLs. Subsequently, we comprehensively investigated the predictive ability, biological properties, immune infiltration, TMB, and drug responsiveness of the 4-DRLs signature. The implications of our findings hold the potential to offer novel perspectives and approaches for clinical immunotherapy strategizing and personalized patient management.

## 2. Results

### 2.1. Characterization of Disulfidptosis-Related Lncrna (DRLs) Based Molecular Subgroups in COAD

Figure 1 illustrated the procedure of assessing the prognostic relevance of DRL expression in colon cancer. Initially, 434 COAD patients with comprehensive clinical data from the TCGA database were randomly allocated into two groups: a training set (N = 217) and a validation set (N = 217). Table 1 presents the clinical characteristics of both sets and reveals no notable variations in clinical traits between the two sets. A total of 20 DRGs identified based on the literature review and previous studies were used to determine DRLs (Figure 2A). A total of 2230 DRLs were identified based on Pearson analysis (|r| > 0.4, *p* < 0.001), and the relation between DRGs and DRLs is presented in Figure 2A. Among the 20 DRGs in this study, CD2AP and MYL6 have the highest number of connections with DRLs according to the Sankey diagram (Figure 2A). To explore the molecular subtypes of COAD based on DRLs, an unsupervised consensus clustering algorithm was performed on the cohort of 434 COAD patients. The resulting heatmap indicated an optimal classification with k = 3, wherein gene cluster 1 comprised 275 samples, gene cluster 2 had 52 samples, and gene cluster 3 included 107 samples (Figure 2B,C). Subsequently, Kaplan–Meier survival analysis revealed distinct clinical prognostic outcomes among the different subgroups. Notably, patients in COAD cluster 3 exhibited significantly lower OS rates compared to those in clusters 1 and 2 (Figure 2D, *p* < 0.01).

### 2.2. Recognition of a Prognostic Disulfidptosis-Related Lncrna Signature

In total, 34 DRLs significantly related to patient OS were obtained through univariate Cox regression analyses (*p* < 0.05, Figure 3A, Appendix A) in the training set. A total of 23 of the 34 DRLs had a hazard ratio (HR) > 1, which indicated that they were bad prognostic predictors, while the remaining 11 DRLs were protective factors with HR < 1. Furthermore, four candidate lncRNAs were ultimately identified through Lasso and the multivariate Cox regression method (Figure 3B–D), which included AC007728.3, AP003555.1, ATP2B1.AS1, and NSMCE1. DT. Consistent with the trend of the univariate Cox regression analysis, AC007728.3 served as a protective indicator, whereas AP003555.1, ATP2B1.AS1, and NSMCE1.DT were classified as risk factors with HR > 1 (Figure 3D). Subsequently, the predictive risk assessment model was developed using multivariate Cox regression relying on the 4 DRLs. The model assigned a risk score to each patient by the given equation: Risk Score = ExpressionAC007728.3 × (−2.2208778) + ExpressionAP003555.1 × 0.3728466 + ExpressionATP2B1.AS1 × 2.942407 + ExpressionNSMCE1.DT × 2.8856887). The relationships between the four DRLs and DRGs are shown in Figure 3E.

### 2.3. The Risk Score Could Be an Independent Prognostic Factor and Assist in Predicting Clinical Outcomes for COAD Patients

The patients were categorized into high- or low-risk groups for further survival analysis based on the median risk scores in each dataset. Figure 4 displayed the risk score and survival status of patients in the training, testing, and combined sets. The results showed a significant correlation between the risk score and the survival of patients in all datasets. Patients in the high-risk group exhibited significantly poorer overall survival in comparison to those with lower risk scores (*p* < 0.001, Figure 4G–I). Moreover, higher mortality was observed as their risk scores increased (Figure 4A–I). Both univariate and multivariate Cox regression analyses indicated that the risk groups, categorized according to the signature based on the four DRLs, were identified as an autonomous prognostic factor for COAD patients in comparison to other clinicopathological characteristics (Figure 4J,K).

### 2.4. Validation of the 4-DRLs Predictive Signature and Construction of A Nomogram Combining Clinical Characteristics

For the 1-, 3-, and 5-year follow-up periods, the ROC curves demonstrated area under the curve (AUC) values of 0.65, 0.73, and 0.66, respectively (Figure 5A). Additionally, an ROC curve was generated to validate the superior prognostic accuracy of the risk score in comparison to alternative clinical variables such as age, gender, stage, tumor (T) size, lymph node (N), and metastasis (M) (Figure 5B). AUC for the risk score was 0.73, indicating stronger predictive capability than other clinical features except T (Figure 5B). The model’s accuracy was verified by calculating the C-index through cross-validation and non-cross-validation, yielding a score of 0.819. Furthermore, the new model was developed incorporating the clinical factors and risk score, and its overall improvement over the old model (excluding the risk score) was assessed using the IDI index, resulting in a value of 0.039 (*p* < 0.01). This indicated that the risk score positively contributed to the constructed model. Consistent with the former results, DCA was utilized to further evaluate the model’s performance, revealing a higher net benefit rate for risks core in both median survival time, and 3-year survival time (Figure 5C,D).

In order to improve the practicality of the signature, we created a predictive nomogram by adding up the assigned scores of relevant clinical factors and risk scores on a points scale (Figure 5E). This allowed for accurate prediction of the probability of survival. As illustrated in Figure 4F, the selected patient’s probability of 1-, 3-, and 5-years OS was determined to be 0.985, 0.961, and 0.924, respectively. Additionally, the consistency of the nomogram predictions and the actual measured outcome were validated by the calibration curves. The findings indicated a strong concordance between clinical outcomes and predictive values, as depicted in Figure 5F. To summarize, these results indicate that the nomogram combined the 4-DRLs predictive signature and clinical features could accurately forecast the clinical prognosis of patients with COAD.

### 2.5. Predicting the Prognosis of High- and Low-Risk-Group Patients with the Clinical Characteristics

Based on the 4-DRLs predictive signature, we compared the survival probabilities of high- and low-risk groups among COAD patients according to the age, gender, stage, and TNM stage. The findings indicated that, in relation to all clinical variables, except T1/T2, the OS in high-risk group was significantly shorter compared to that of the low-risk group (Figure 6A–L). A possible explanation is that poor prognosis in advanced-stage COAD led to a relatively smaller number of T1/T2 patients, which might cause some degree of error in the results. Overall, these results suggest that the 4-DRLs predictive model has immense potential for predicting COAD prognosis and can be applied across various clinical variables.

### 2.6. Biological Functional Analysis by GO, KEGG, and GSEA Analysis

PCA was employed to visually display variations in distribution between groups at high and low risk (Figure 7A–D). The results showed that there were no notable disparities in the expression patterns of total genes, DRGs, or DRLs among the two risk groups (Figure 7A–C). However, the four-DRLs used in the predictive model exhibited the highest discriminatory power in distinguishing between high- and low-risk patients (Figure 7D). Subsequently, we identified 1466 differentially expressed genes (DEGs) by comparing the mean expression values of two group (Padjust < 0.05, |log2 (FC)| > 0.7), as indicated by the MA plot (Figure 7E).

To examine the biological characteristics of DEGs, we conducted GO and KEGG enrichment. According to the biological process (BP), the DEGs played a significant role in the “regulation of membrane potential”, “modulation of chemical synaptic transmission”, “regulation of trans-synaptic signaling”, and “muscle contraction”. Within the realm of cellular components (CC), “neuronal cell body”, “perikaryon” and “transmembrane transporter complex” were significantly abundant. Furthermore, molecular functions (MFs) analysis revealed a significant enrichment of DEGs in “signaling receptor activator activity”, “receptor ligand activity”, and “channel activity”. (Figure 7F). These findings indicated that DEGs were involved in metabolism-related biological functions. The KEGG results were consistent with those of GO analysis, revealing significant pathway enrichments in “neuroactive ligand-receptor interaction”, “calcium signaling pathway”, “bile secretion”, “insulin secretion”, and “tyrosine metabolism” (Figure 7G,H). These pathways are also related to cellular signaling and metabolism. Furthermore, a significant enrichment of pathways associated with poor tumor prognosis was observed in the high-risk group through the GSEA analysis, such as “E2F targets pathway”, “G2M checkpoint pathway”, “MTORC1 signaling pathway”, “MYC targets pathway”, and “RIBOSOME” (Figure 7I,J).

### 2.7. Tumor Immune Microenvironment Landscape of COAD Patients Based on Prognostic Signature

To explore the correlation between the predictive signature of 4-DRLs and the immune process in patients with COAD, we characterized the landscape of tumor immune cell infiltration for all patients in the TCGA database using different algorithms, such as TIMER, CIBERSORT, CIBERSORT-ABS, QUANTISEQ, MCPCOUNTER, XCELL, and EPIC (Figure 8A). Several major algorithms determined that there were notable variations in immune cells between the two groups, as depicted in Figure 8B–F. In TIMER (Figure 8B), the high-risk group exhibited elevated expression levels of anti-tumor immune cells, encompassing CD8+ T cells, CD4+ T cells, B cells, macrophages, and dendritic cells. The CIBERSORT analysis showed notable rises in activated mast cells and activated NK cells, while resting mast cells and mast cell NK cells decreased (Figure 8C). Additionally, XCELL results demonstrated a notable decrease in CD4+ Th1 cells, a key cell for generating long-lasting anti-tumor immune responses, in the high-risk group, which may be a possible factor for the poorer prognosis (Figure 8D). Finally, the results from both MCPCOUNTER (Figure 8E) and EPIC (Figure 8F) collectively demonstrated that cancer-associated fibroblasts, crucial constituents of the tumor mesenchyme, exhibited significantly higher levels in the high-risk group. Altogether, these findings demonstrated a robust correlation between risk score and the infiltration of immune cells, as observed in all of these algorithms.

Based on the TIDE analysis, patients at high risk showed a significantly lower score in comparison to those at low risk (*p* < 0.001, Figure 9A). This suggested that high-risk patients may benefit more from immunotherapy due to a reduced likelihood of immune evasion. Furthermore, we assessed the discrepancies in immune-related functions between the high-risk and low-risk cohorts. The high-risk sets showed a significant enrichment of “type_II_IFN_Reponse” as illustrated in Figure 9B. Finally, we performed a comparison between the two groups to assess the levels of immune checkpoint genes expression. According to Figure 9C, there were notable variations in 14 checkpoint genes between the two groups. Significantly, out of these genes, 12 exhibited increased expression in the high-risk category, including CD28, TIGHT, and PD-L2. In contrast, the risk score showed a negative correlation only with PVR and CEACAM1. These immune checkpoints collectively regulate T cell activation and function, some of which are critical for immune tolerance and autoimmunity. These findings provided a potential explanation for the poorer OS observed in patients with higher risk scores. On the other hand, it also implied that these patients might have increased immune responsiveness and could potentially experience enhanced benefits from immune checkpoint inhibitor (ICI) therapies.

### 2.8. Tumor Mutation Burden (TMB) Characteristic and Drug Sensitivity in the 4-Drls Predictive Signature

To investigate the disparities in cancer-associated gene mutations between the high- and low-risk groups, we acquired somatic mutation data from the TCGA database. The examination uncovered the 15 genes that were mutated most frequently, comprising APC, TP53, TTN, KRAS, SYNE1, MUC16, PIK3CA, FAT4, RYR2, CSMD3, ZFHX4, DNAH5, OBSCN, LRP1B, and PCLO (Figure 10A). Although the TMB did not differ significantly in an overall view between the two groups (Figure 10B), incorporating TMB and risk score for grouping analysis could have effectively predicted patient prognosis, as shown in Figure 10C,D. The high TMB and high-risk group exhibited the poorest prognosis compared to other groups, whereas the low TMB and low-risk group demonstrated the longest survival time (Figure 10C,D).

To evaluate the clinical utility of the 4-DRLs predictive signature and search the potentially effective drugs, we analyzed the chemotherapeutic drug by using “oncoPredict” packages. Following the prediction of sensitivity for 743 compounds across all patients, we performed a Wilcoxon test to compare differences between the two risk groups (adjust *p* < 0.01). A total of 224 of the 545 drugs in CTRP and 31 of the 198 drugs in GDSC were identified with a significantly difference between high- and low- risk groups (Figure 11A). Lapatinib, Gemcitabine, and MK-2206 were identified as promising compounds at the intersection of the two databases (Figure 11B–D). The compounds exhibited a greater IC50 in the high-risk category, which could potentially be more suitable for patients in the low-risk category (Figure 11C,D). Figure 11E–H display several chemotherapy drugs widely used in clinical for colorectal cancer, with 5-fluorouracil and oxaliplatin exhibiting lower IC50 values in the low-risk group, rendering them more suitable for patients in this category (Figure 11E,F). Conversely, dabrafenib and temozolomide showed lower IC50 in the high-risk category, indicating their sensitivity to patients classified as high-risk (Figure 11G,H).

### 2.9. External Datasets Validation of the Prognostic Ability of the 4-Drls Predictive Signature

To further confirm the accuracy of the signature in predicting patient prognosis, we used the Kaplan–Meier plotter database to examine the predictive significance of ATP2B1.AS1 and NSMCE1.DT in colon cancer, while AP003555.1 and AC007728.3 were absent in the database. Patients who exhibited elevated levels of ATP2B1.AS1 experienced significantly reduced PPS and RFS (HR > 1, *p* < 0.01, Figure 12A–C). Conversely, patients with increased expression of NSMCE1.DT demonstrated noticeably shorter OS and RFS (HR > 1, *p* < 0.05, Figure 12D–F). The results align with prior studies that recognized ATP2B1.AS1 and NSMCE1.DT as elements that contribute to patient prognosis. In summary, external database analysis provided clear validation for the ability of the 4-DRLs signature to evaluate the prognosis of COAD patients.

### 2.10. Validation of 4-DRLs Expression In Vitro Experiments

To further evaluate the expression of these four DRLs, RT-qPCR was performed both in the cell lines and samples from colon cancer patients. We conducted cell line screening in the CCLE database and chose LoVo and HCT116 cells based on their comprehensive expression levels of the four DRLs (Appendix A). As shown in Figure 13A, compared to normal human colonic cells (NCM460), the expression of ATP2B1.AS1 and NSMCE1.DT were higher in colonic cancer cells (including HCT116 and LoVo), while AC007728.3 exhibited a contrasting trend. AP003555.1 did not exhibit significant differences between cell types. We also investigated the expression levels of these four lncRNAs in carcinoma and adjacent tissues obtained from colon cancer patients. Consistent expression trends were observed, with ATP2B1.AS1, AP003555.1, and NSMCE1.DT significantly overexpressed in tumor tissues (Figure 13B–E). In contrast, AC007728.3 displayed reduced expression levels in tumor tissues as compared to paracancerous tissues (Figure 13C). These results serve to corroborate the accuracy of the above bioinformatics analysis, while also providing further validation for the clinical significance of the predictive model.

## 3. Discussion

In 2020, colon cancer ranked as the third most widespread cancer worldwide in terms of occurrence and the second primary reason for cancer-related fatalities [1], highlighting the urgency for early and accurate identification of high-risk patient subgroups to enable tailored treatment. The mainstream classification methods for colon cancer mainly relied on TNM staging system. Although this method was helpful in selecting appropriate treatments, patients within the same subtype could still present with varying clinical outcomes. Therefore, more precise classification strategies were needed to facilitate personalized therapeutic determination.

Up to now, there are several types of RCD pathways with unique characteristics and underlying mechanisms that have been extensively studied, such as apoptosis, necroptosis, pyroptosis, ferroptosis, and cuproptosis. While each of these RCD pathways has distinct features, they can also be interconnected through various signaling pathways. For instance, studies have shown cross-talk between autophagy and apoptosis, as well as between necroptosis and ferroptosis [13,14]. A recent study demonstrated that high expression of SLC7A11 can expedite the depletion of NADPH within the cytoplasm, particularly under conditions of glucose starvation, which may inhibit ferroptosis and induce a new form of cell death, disulfidptosis [5]. The recognition of this mechanism holds the potential to foster the advancement of efficacious cancer treatments. However, there is currently limited research on the application of disulfidptosis in colon cancer. Therefore, further studies on disulfidptosis are urgently needed to deepen our understanding of its potential application in cancer therapy.

Several research studies have documented the crucial function of long non-coding RNAs (lncRNAs) in the advancement of colon cancer, and multiple lncRNA signatures associated with RCD have been discovered for prognostic prediction [11,15,16]. However, no investigation has been conducted on the association between DRLs and colon cancer. Here, according to the expression level of DRLs in COAD samples, we identified three robustly distinct disulfidptosis-related molecular subtypes, Cluster 1, Cluster 2, and Cluster 3. These LYAG subtypes had significant differences in prognosis. However, due to the existence of many other factors that also influence tumor infiltration and staging, such as colon location and microsatellite instability (MSI), it is essential to consider them in subsequent research studies in order to obtain more reliable results. Then, we established a risk model for colon cancer prognosis by DRLs. Firstly, 34 DRLs associated with prognosis were identified by univariate regression analysis. Subsequently, LASSO and multivariate Cox regression analyses were employed to screen and identify four lncRNAs (AC007728.3, AP003555.1, ATP2B1.AS1, and NSMCE1.DT) that were introduced into the model. Among them, AP003555.1, ATP2B1.AS1, and NSMCE1.DT have been shown in previous studies to be closely related to colon cancer prognosis. In particular, AP003555.1 was reported to be an effective prognostic element in signatures based on ferroptosis-related lncRNAs, which also indirectly suggests the strong relationship between ferroptosis and disulfidptosis [14,15]. ATP2B1.AS1 is strongly associated with cell inflammation and can regulate the miR-23a-3p/TLR4 axis, exacerbating sepsis-induced cell apoptosis and inflammation [17]. Furthermore, it has been found to promote cerebral ischemia/reperfusion injury and worsen myocardial infarction by activating the NF-κB signaling pathway [18,19]. As a lncRNA related to pyroptosis, it also served as a risk factor and predictor of prognosis in gastric cancer and colon cancer patients [20,21]. In further research, NSMCE1.DT was found to be highly expressed in colon cancer patients as a lipid metabolism-related lncRNA [22]. These results are consistent with our findings, indicating that AP003555.1, ATP2B1.AS1, and NSMCE1.DT may be risk indicators for colon cancer patients with HR > 1. ATP2B1.AS1 and NSMCE1.DT were also validated in external datasets and exhibited a correlation between their high expression and poorer prognosis (Figure 12). AC007728.3 has not been characterized in any studies.

Based on these four lncRNAs, we established a new clinical prognosis model that is more suitable for clinical application than some signatures that have been identified. This signature includes only four lncRNAs, making it convenient and less time consuming to detect. In our study, we employed a random division of the TCGA-COAD cohort into training and testing subsets. Subsequently, patients were categorized into high- and low-risk groups based on their respective risk scores calculated using the developed model. The high-risk group showed poorer prognosis in terms of survival curves (Figure 4), which was consistent with clinical subgroup analyses (except for T1/2 stages) (Figure 6). One plausible explanation for this observation could be that fewer patients were diagnosed at the T1-2 stages due to the typically poorer prognosis associated with advanced COAD. Compared with traditional TNM stage and other clinical pathological features, the risk score demonstrated more outstanding predictive ability for the prognosis of colon cancer patients. ROC curves, C-index, IDI index, and DCA all confirmed this fact, in addition to Kaplan–Meier survival curves. The IDI and DCA provided a comprehensive evaluation of the model performance, both indicating that the new model with the addition of risk score has better overall predictive ability than the old one. Additionally, a nomogram incorporated clinical variables along with the risk score, which demonstrated superior predictive ability compared to the existing clinical staging system. The risk score exhibited independence from significant prognostic factors in colon cancer, and we used the median value to categorize patients into different groups. Utilizing the median value for classification purposes is regarded as a more practical and objective approach, especially when compared to optimal cutoff values, which may only perform well in specific cohorts and lack universality.

The PCA results showed that the 4 DRLs had excellent discriminatory ability between low- and high-risk patients. To further understand their biological properties, we carried out GO and KEGG analyses. The GO analysis revealed that DEGs were mainly related to cellular signaling and muscle contraction, indicating a close association with cellular metabolism. This is consistent with the fact that disulfidptosis is triggered by disulfide bond formation in protein molecules due to NADPH depletion and disulfide stress, both of which are closely linked to energy supply and cellular metabolism [5]. KEGG analysis indicated that the “neuroactive ligand-receptor interaction” and “calcium signaling pathway” were significantly enriched. Although neuroactive ligand–receptor interaction is primarily associated with neurological conditions, there is increasing evidence to suggest its involvement in cancer progression and metabolism. Research has revealed that dysregulated expression of genes involved in neurotransmitter signaling within colorectal cancer contributes to the facilitation of tumor growth and metastasis. This phenomenon occurs through the stimulation of cell proliferation, migration, invasion, and angiogenesis processes by neurotransmitters [23]. The mechanism of Ca^2+^ channels acting on tumor is complex and can affect tumor progression from multiple aspects. It has been reported that Ca^2+^ could activate NF-κB, NFAT, and CREB pathways, thereby playing an important role in tumor immunity cells and progression [24,25]. Moreover, the high-risk group exhibited a significant enrichment of pathways associated with poor tumor prognosis, including “E2F targets pathway”, “G2M checkpoint pathway”, “MTORC1 signaling pathway”, “MYC targets pathway”, and “RIBOSOME” based on GSEA analysis (Figure 7I,J) [26,27,28,29,30]. In the field of precision oncology, the strategy of targeting cancer metabolism to selectively eradicate tumor cells has gained significant traction and widespread adoption [31]. However, it is worth noting that disulfidptosis was also closely associated with immune cells and functions according to our immune analysis results, while the metabolic treatment of cancer cells may have an impact on non-cancerous cells in the meantime, particularly immune cells. This effect could potentially limit the therapeutic efficacy of cancer metabolic therapy, which emphasizes the need for a comprehensive evaluation of its effectiveness and potential side effects before implementing such therapies [31].

Recent studies have suggested a close association between disulfidptosis and immune infiltration, with high disulfidptosis subtypes exhibiting higher immune scores [6]. Our findings are consistent with these results, as the estimated immune cell infiltration via seven different algorithms revealed that high-risk patients exhibited a propensity for elevated antitumor immune activity. Specifically, we observed higher expression levels of major anti-tumor immune cells such as CD8+ T cells, CD4+ T cells, and macrophages in the high-risk group by TIMER analysis (Figure 8B). Our findings challenge previous perceptions that the high degree of CD8+ T cell infiltration generally indicates a better survival prognosis [32]. There are several studies also reported that elevated levels of CD8+ T cells were sometimes linked to shorter survival time [33,34]. Moreover, XCELL analysis revealed a significant reduction in Th1 cells—crucial cells for generating long-lasting anti-tumor immune responses—in the high-risk group [35], which is likely to contribute to the group’s poorer prognosis (Figure 8D). The tumor immune environment is a complex setting in which not only immune cells, but also various factors such as immune checkpoints, regulatory cells, inflammatory cytokines, and the tumor microenvironment can all affect immune function. According to TIDE analysis, high-risk patients exhibited a low probability of immune escape (Figure 9A). This suggests that those patients may benefit more from immunotherapy and potentially experience less resistance to ICI. In addition, in the high-risk group, there was a notable elevation in the expression levels of various immune checkpoints, including CD28, TIGHT, and PD-L2. Therefore, it can be inferred that these patients may exhibit heightened immunoreactivity and potentially derive greater benefit from ICI therapies [36]. However, further research is needed to explore whether inhibitors targeting these checkpoints represent promising antitumor agents for colon cancer. At present, the only ICI approved for the treatment of colon cancer are PD-1/PD-L1 inhibitors and CTLA-4 inhibitors. CD28, TIGHT, and CTLA-4 belong to a family of immunoglobulin-related receptors that regulate different aspects of T-cell immunity [37]. As a negative costimulatory molecule, PD-L2 can suppress T-cell activation and function by binding with PD-1, contributing to immune tolerance and immune evasion [38]. Therefore, all of these immune checkpoints represent promising targets for the treatment of colon cancer.

TMB demonstrates a strong correlation with tumor immune response and prognosis. Previous studies have reported that TMB can predict the response to ICI in metastatic colorectal cancer patients with high microsatellite instability (MSI-high) [39]. Despite the lack of a substantial statistical disparity in TMB between the high- and low-risk cohorts within our investigation, patients presenting both elevated TMB and high-risk factors demonstrated the most unfavorable prognosis when contrasted with the remaining patient populations. The findings additionally confirmed the specific association between TMB and the prognosis and survival of tumors. Furthermore, the risk groups determined by the 4-DRLs signature also displayed distinct drug sensitivities. Lapatinib, Gemcitabine, and MK-2206 were identified as promising compounds at the intersection of the GDSC and CTRP databases and were more appropriate for patients in the low-risk group (Figure 11C,D). Lapatinib is an approved targeted therapy acting as a dual inhibitor of HER2/EGFR for metastatic breast cancer [40]. Interestingly, HER2 activating mutation has emerged as a significant target for the treatment of colon cancer [41]. Gemcitabine and MK-2206 have also been extensively investigated as potential chemotherapy agents for the treatment of colon cancer [42,43]. In addition, colon cancer has long been treated with a combination of 5-fluorouracil and oxaliplatin, which have been approved as primary treatment options [44]. The low-risk group demonstrated lower IC50 values for the aforementioned drugs, indicating their potential suitability for patients in this category. Conversely, dabrafenib and temozolomide are better options for patients in the high-risk category due to their notably lower IC50 values. These agents were also recognized as highly promising chemotherapy regimens and widely utilized in clinical practice [45,46]. The findings from our study may provide guidance in selecting appropriate drugs for patients with varying risk scores in the future.

Finally, we conducted in vitro experiments to confirm the expression level of these four lncRNAs. The expression patterns closely aligned with the predictions derived from our earlier bioinformatic analysis. Specifically, ATP2B1.AS1 and NSMCE1.DT exhibited higher expression in colon cancer tissues compared to noncancerous tissue, as well as higher expression in colonic cancer cell lines than in human normal cell lines. On the other hand, AC007728.3 displayed an inverse pattern, aligning with our findings. The expression difference of AP003555.1 in cells did not reach statistical significance, possibly attributable to variations among different cell lines.

There are several limitations that need to be addressed in our study. Firstly, due to the novelty of disulfidptosis as a newly discovered form of regulated cell death (RCD), there is limited research and databases available on this topic. Therefore, we only included 20 DRGs based on a limited number of research findings, which may have affected the comprehensiveness of the results presented in the paper. In future studies, we will comprehensively summarize and analyze all the published literature and relevant databases to provide a more comprehensive and accurate analysis. Secondly, due to the limited research on the four selected lncRNAs, we were unable to obtain comprehensive lncRNA annotations and clinical information from databases such as GEO, ICGC, and others. This limitation highlights the ongoing importance of lncRNAs, which remain partially obscured by the constraints of current technology. Although two of the lncRNAs were confirmed through the external Kaplan–Meier plotter database, which includes GEO, EGA, and TCGA data, further validation of the risk signature necessitates independent colon cancer cohorts to strengthen its credibility and robustness. Thirdly, although we performed qRT–PCR to examine the expression levels of the four DRLs in several clinical samples and two colon cancer cell lines, the limited sample sizes used in our study posed a constraint. Conducting larger-scale studies with sufficient sample sizes would provide stronger evidence and validate the prognostic significance of the model. Lastly, the mechanisms through which these lncRNAs affect the immune landscape and drug sensitivity are still unknown. Additional extensive investigation is required to examine the complex association between these lncRNAs and DRGs, aiming to reveal possible targets for successful therapeutic approaches.

For future research, it is crucial to validate our risk prediction model using external independent patient cohorts to establish its generalizability and clinical utility. Mechanistic investigations also should be conducted to uncover the biological pathways through which disulfidptosis lncRNA influences colon cancer. Moreover, integrating multiple predictive factors, such as clinical indicators and genetic variations, into our risk prediction model is essential. This comprehensive approach has the potential to improve the accuracy and applicability of the model in real-world clinical settings.

## 4. Materials and Methods

### 4.1. Data Acquisition

The RNA-Seq data and corresponding clinical information of COAD were downloaded from TCGA database (https://portal.gdc.cancer.gov/repository (accessed on 15 May 2023)), which included 434 COAD tumors and 41 normal samples. “limma” script [47,48] was used to convert the transcriptome matrix of TCGA-COAD fragments per kilobase million (FPKM) to transcripts per million (TPM), to facilitate further analysis.

### 4.2. Identification the Expression Matrix of DRLs and Molecular Subtype Characterization

The list of 18 disulfidptosis-related genes (DRGs) were obtained from previously published studies [5]. The DRGs and DRLs expression matrix was retrieved by “limma” packages [47,48]. The criteria for DRLs were |Pearson R| > 0.5 and *p* < 0.001. Then, the “ggplot2” and “ggalluvial” scripts were utilized to draw the Sankey diagram [49]. The “ConsensusClusterPlus” script was used to categorize COAD samples into distinct molecular subtypes by analyzing the expression levels of DRLs, employing 1000 iterations and an optimal classification range of K = 2–5 [50]. Subsequently, we assessed the clinical survival outcomes of COAD in relation to these molecular subtypes based on the “survival” script.

### 4.3. Construction and Validation of Prognostic Signature

By “caret” script, patients with COAD were randomly divided into training and testing groups in a 1:1 ratio [51]. The prognostic model was constructed using the training set, and validation was performed using the test set and entire set. DLRs with *p* < 0.05 that resulted from univariate Cox regression were kept for the subsequent stage. Further, a total of 4 prognostic CDRLs were obtained by the least absolute shrinkage and selection operator (LASSO) and multivariate Cox regression analysis. Afterwards, we developed the prognostic model utilizing the four DRLs, and computed a risk score for each patient using the provided equation:(1)$$Risk\;score=\sum_{n=1}^{i}\left({Coef(lncRNA}_{i})\;*\;{Expr(lncRNA}_{i})\right)$$

The calculation of ***Coef*** (***lncRNA_i_***) is performed using multivariate Cox regression analysis. Patients in the training set, test set, and entire set were divided into low- and high-risk groups separately, based on the median risk score. In three sets, the K–M analysis was performed to forecast the overall survival (OS) among the high/low-risk group. Furthermore, the model’s accuracy was assessed using various methods including the receiver operating characteristic curve (ROC), C-index, Integrated Discrimination Improvement (IDI) index, decision curve analysis (DCA), nomograms, and calibration curves. These above series of processes were performed using R packages in software R 4.2.1 mainly including “survival”, “glmnet”, “survminer”, “timeROC”, “ggDCA”, “rms”, and “forestplot” [52,53].

### 4.4. Functional Enrichment Analysis

PCA was employed to visually display the distinctive function of the signature with packages of “scatterplot3d” and “limma”. The genes differentially expressed between the two groups were identified (*Padjust* < 0.05, |log2 (fold change)| > 0.7), and visualized by MAplot under the “limma”, “egdeR”, “ggplot2”, and “ggrepel” packages [47,54,55]. Gene Ontology (GO) and the Kyoto Encyclopedia of Genes and Genomes (KEGG) and Gene Set Enrichment Analysis (GSEA) analysis were performed with packages “clusterProfiler” [56].

### 4.5. Assessment of Immune-Infiltration Characteristics

The extent of immune cell infiltration in the two groups was assessed using seven different algorithms, including TIMER, CIBERSORT, CIBERSORT-ABS, QUANTISEQ, MCPCOUNTER, XCELL, and EPIC [57,58,59,60,61]. Tumor Immune Dysfunction and Exclusion (TIDE: http://tide.dfci.harvard.edu/ (accessed on 15 May 2023)) algorithms were used to assess the potential efficacy of tumor immunotherapy [62]. Heat map, boxplot, or violin plot were plotted to exhibit the difference expression of immune cell infiltration, TIDE, immune function, and immune checkpoints by packages “limma”, “GSVA”, “GSEABase”, “ggplot2”, “ggpubr” and “ggExtra” [63].

### 4.6. TMB and Drug Sensitivity Analysis

The TMB of COAD patients in various groups were identified by package “maftools” [64]. The drug sensitivity data were download from the Genomics of Drug Sensitivity in Cancer (GDSC) [65] and the Cancer Therapeutics Response Portal (CTRP) database [66]. “Oncopredict” package was used to determine the half-maximal inhibitory concentration (IC50) of drugs in two sets [67].

### 4.7. External Dataset Validation

Kaplan–Meier plotter database was consumed to predict the OS, progression-free survival (PPS) and recurrence-free survival (RFS) of patients in different expression levels of DLRs [68].

### 4.8. Cell Culture

The cell line LncRNA expression matrix of COAD was obtained from the CCLE dataset (https://portals.broadinstitute.org/ccle (accessed on 15 May 2023)) [69]. Normal colon mucosal epithelial cell (NCM460) was purchased from BNCC Company (Beijing, China) and human colonic cancer cells, including LoVo and HCT116 were purchases from American Type Culture Collection (ATCC, Manassas, VA, USA). These cells were cultured in complete DMEM, F-12 or McCoy’s 5A medium with 10% fetal bovine serum (Gibco, Waltham, MA, USA), and passaged in a humidified atmosphere containing 5% CO_2_ at 37 °C.

### 4.9. Tissue Sample Collection

All samples were obtained from the Department of Gastroenterology and Hepatology of Tianjin Medical University General Hospital, which was approved by the Medical Ethics Committee of the hospital (Ethical No. IRB2023-W7-107). Prior to data collection, we obtained informed consent from every patient involved in the study. Seven samples from colon cancer patients containing tumor samples and pericarcinous samples between October 2022 and April 2023 were cryopreserved at −80 °C.

### 4.10. RNA Extraction and RT-qPCR

The total RNA in cells and tissues were extracted with Trizol reagent (Vazyme, Nanjing, China, R411-01) and reverse-transcribed using the HiScript III RT SuperMix (Vazyme, China, R323). Quantitative PCR analysis was performed using Universal SYBR Green Fast qPCR Mix (ABclonal, Hong Kong, China, RK21203), and the results were calculated with 2^(−ΔΔCt)^ method with the GADPH serving as the internal control reference. The primer sequences are listed in Appendix A.

### 4.11. Statistical Analysis

Data processing and visualization were performed using R software (version 4.3) and GraphPad Prism 9.0. The correlation between DRGs and DRLs was assessed using Pearson and Spearman correlation analysis. The Wilcox test was used to compare the differences between two groups. Classified variables were analyzed for differences in proportions using the chi-squared test (χ^2^) or Fisher’s exact test. Statistical significance was defined as *p* < 0.05, with all *p*-values considered two-tailed.

## 5. Conclusions

In conclusion, our study pioneers the exploration of DRLs in colon cancer and their clinical significance. Through the development and validation of a novel signature comprising four DRLs, we demonstrated its accurate predictive role in assessing prognosis, immunotherapy response, and chemotherapy sensitivity in colon cancer patients. This research endeavor holds promise in providing innovative insights for predicting clinical outcomes in colon cancer patients. Moreover, it contributed to the advancement of the theoretical foundation necessary for enhancing immunotherapy and personalized anti-tumor treatments.

## Figures and Tables

**Figure 1 ijms-24-12915-f001:**
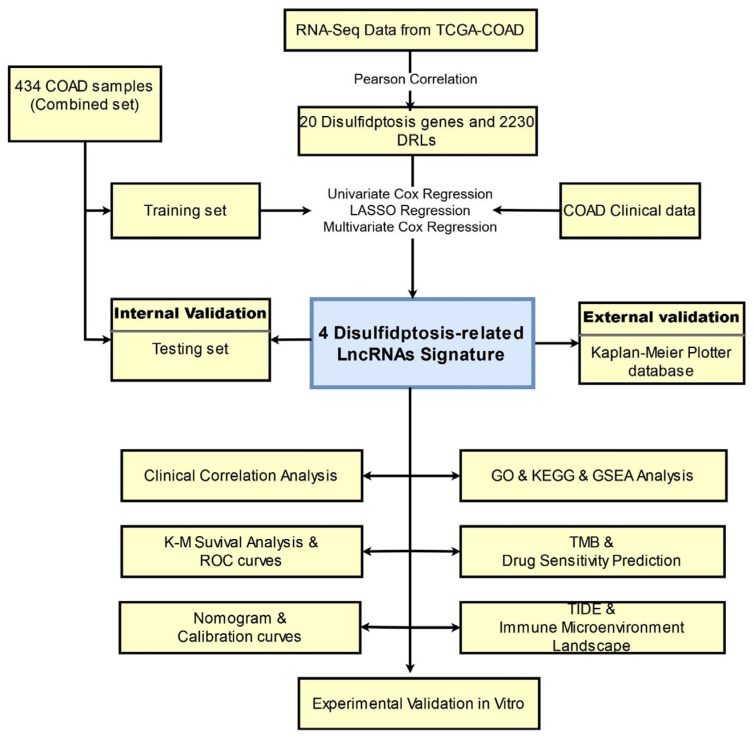
Flow chart of this study.

**Figure 2 ijms-24-12915-f002:**
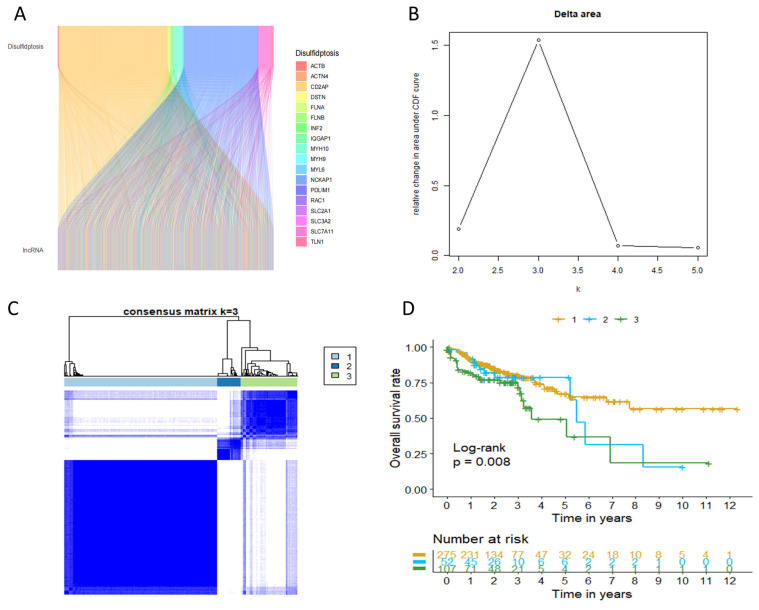
Characterization of disulfidptosis-related Lncrna (DRLs)-based molecular subgroups in COAD. (**A**) The Sankey relation between disulfidptosis-related genes (DRGs) and DRLs; (**B**) the relative change in area under the CDF curve for k  =  2 to k  =  5. (**C**) Unsupervised consensus clustering analysis for COAD. (**D**) Kaplan–Meier curve for three clusters of COAD.

**Figure 3 ijms-24-12915-f003:**
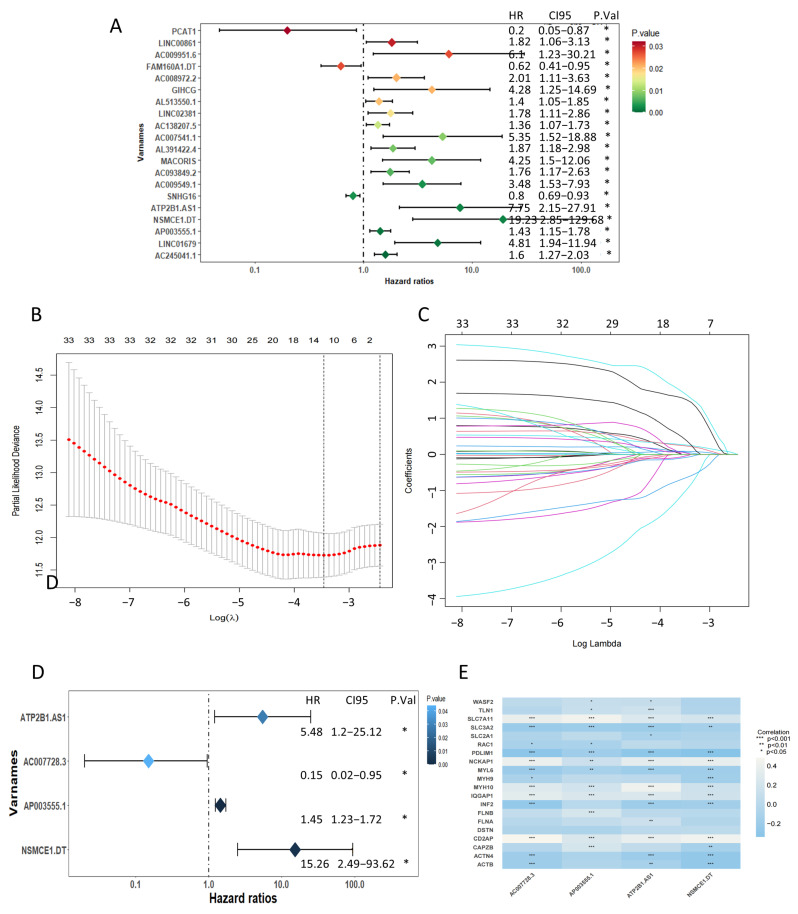
Identification of the prognostic features of colon adenocarcinoma (COAD) linked to disulfidptosis-related lncRNA (DRLs). (**A**) Univariate cox forest map showing the top 20 prognostic DRLs. (**B**) Least absolute shrinkage and selection operator (LASSO) coefficients of DRLs. (**C**) Cross-validation of DRLs in the LASSO regression. (**D**) Multivariate Cox forest map showing the four prognostic DRLs. (**E**) The relationships between the four DRLs and DRGs. *, *p* < 0.05; **, *p* < 0.01; ***, *p* < 0.001.

**Figure 4 ijms-24-12915-f004:**
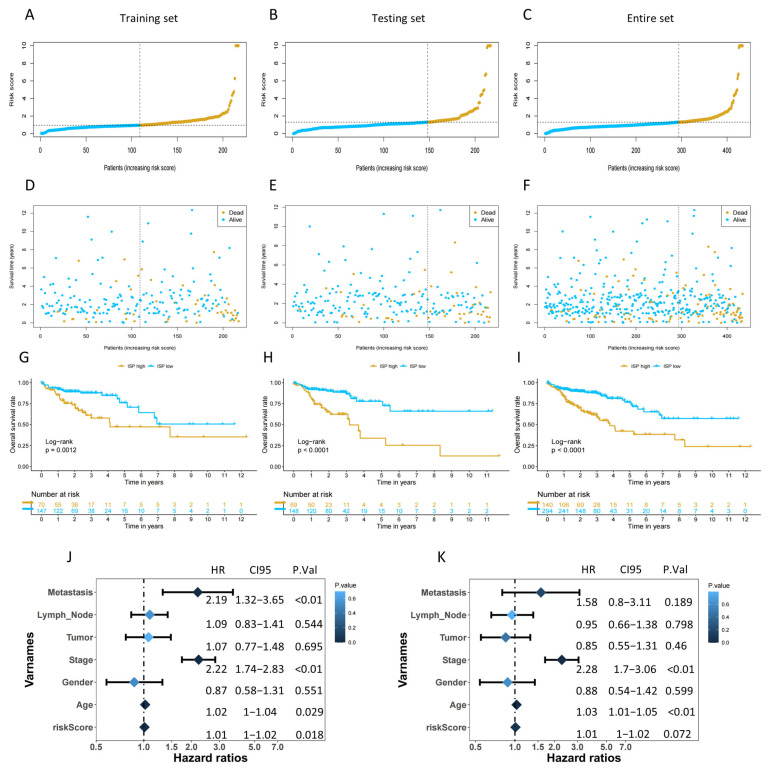
Evaluation and validation of the independent prognostic ability of 4-DRLs signature model in training, testing, and entire sets. (**A**–**C**) The distribution of patient with increasing risk scores. (**D**–**F**) The survival time of patients and risk scores. (**G**–**I**) The Kaplan–Meier (K-M) survival analysis of survival status and overall survival (OS) of COAD patients between high- and low-risk groups. (**J**) A univariate Cox regression analysis of clinical variables and risk score. (**K**) A multivariate Cox regression analysis of clinical variables and risk score.

**Figure 5 ijms-24-12915-f005:**
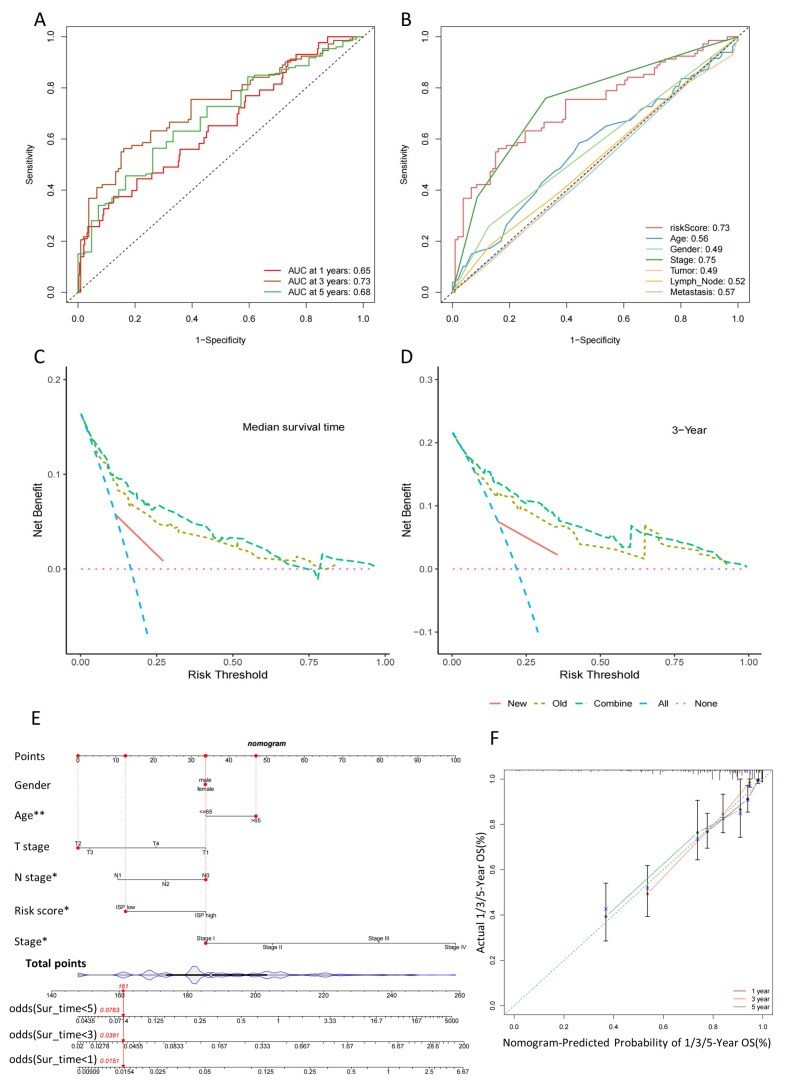
Validation of the predictive model and construction of a nomogram combining clinical characteristics. (**A**) The receiver operating characteristic (ROC) curves show the predictive accuracy in 1-, 3-, and 5-year of the predictive risk model. (**B**) The ROC curves show the predictive accuracy of the predictive risk model and clinicopathological characteristics. (**C**) Decision curve analysis (DCA) shows the overall improvement of the predictive risk model in median survival time. (**D**) DCA shows the overall improvement of the predictive risk model in 3-year survival time. (**E**) The nomogram to predict the 1-, 3-, and 5-year overall survival (OS) rate of colon cancer patients. (**F**) The calibration curve for evaluating the accuracy of the nomogram model in 1-, 3-, and 5-year categories. *, *p* < 0.05; **, *p* < 0.01.

**Figure 6 ijms-24-12915-f006:**
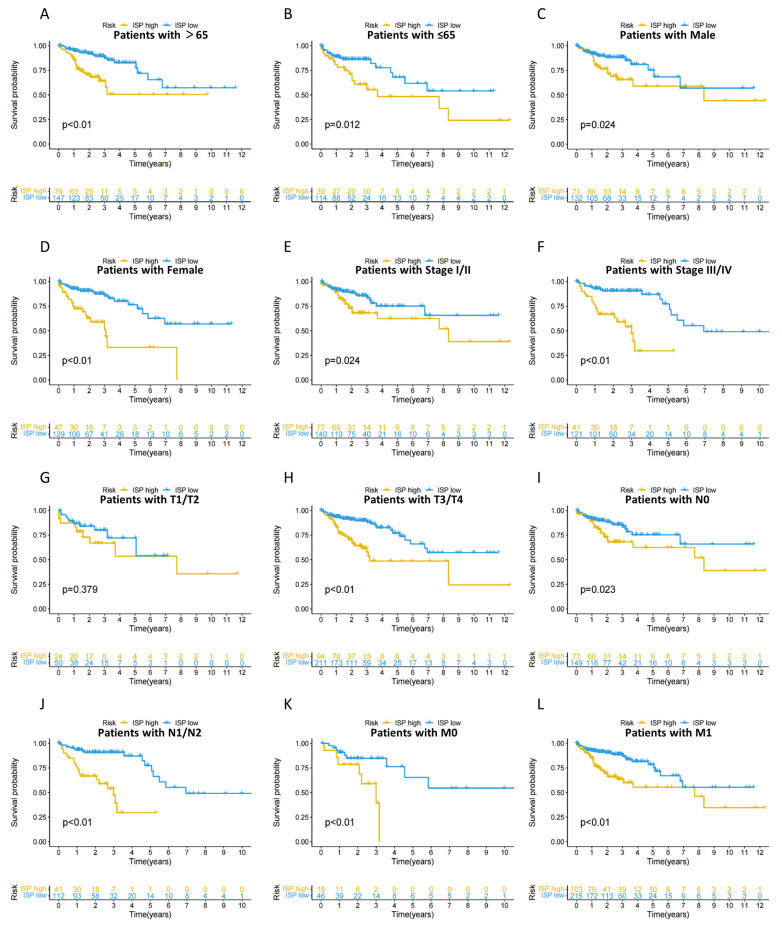
The Kaplan–Meier (KM) survival analysis of low- and high-risk patients with different clinical variables. (**A**,**B**) Age (>65, ≤65); (**C**,**D**) Gender (Male, Female); (**E**,**F**) Stage (Stage I/II, Stage III/IV); (**G**,**H**) T stage (T1/T2, T3/T4); (**I**,**J**) N stage (N0, N1/N2); (**K**,**L**) M stage (M0/M1).

**Figure 7 ijms-24-12915-f007:**
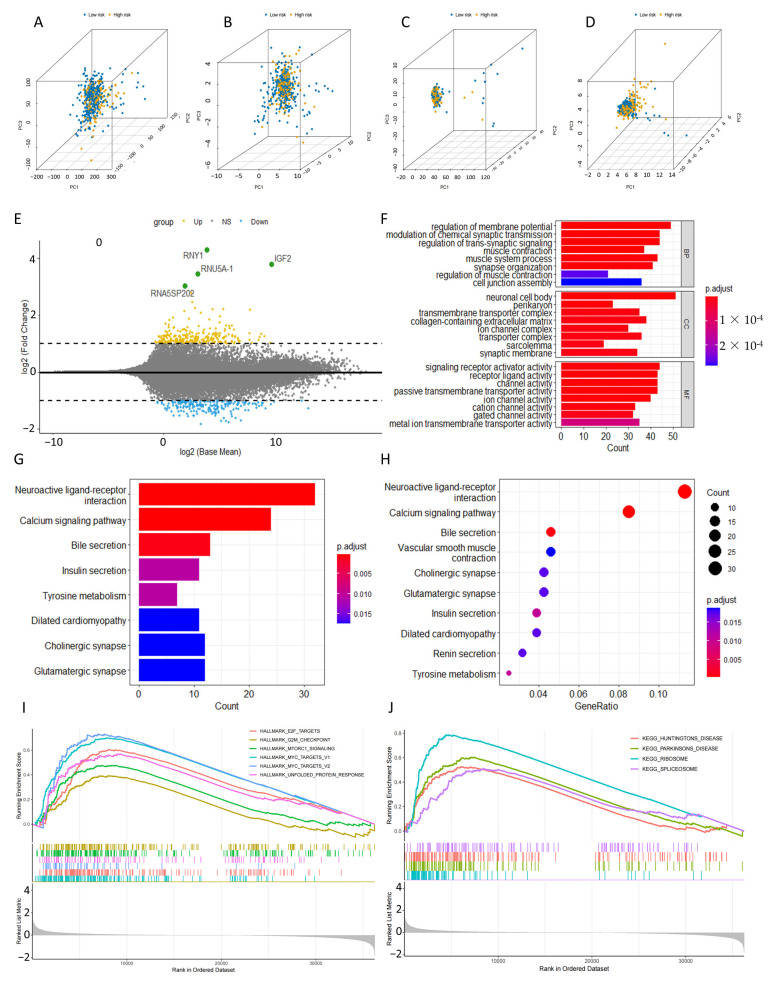
Biological functional and pathway enrichment analysis of the disulfidptosis-related lncRNA prognostic signature. (**A**) PCA about total genes of patients in high- and low-risk groups. (**B**) PCA about disulfidptosis-related genes (DRGs) of patients in high- and low-risk groups. (**C**) PCA about disulfidptosis-related lncRNAs (DRLs) of patients in high- and low-risk groups. (**D**) PCA about the four DRLs used in the predictive model of patients in high- and low-risk groups. (**E**) The MAplot about differentially expressed genes (DEGs) based on the average expression values of high- and low-risk groups. (**F**) GO analysis reveals the diversity of molecular biological processes (BPs), cellular components (CCs), and molecular functions (MFs). (**G**,**H**) KEGG analysis shows the significantly enriched pathways. (**I**,**J**) GSEA demonstrate the enriched pathways in high- and low-risk sets.

**Figure 8 ijms-24-12915-f008:**
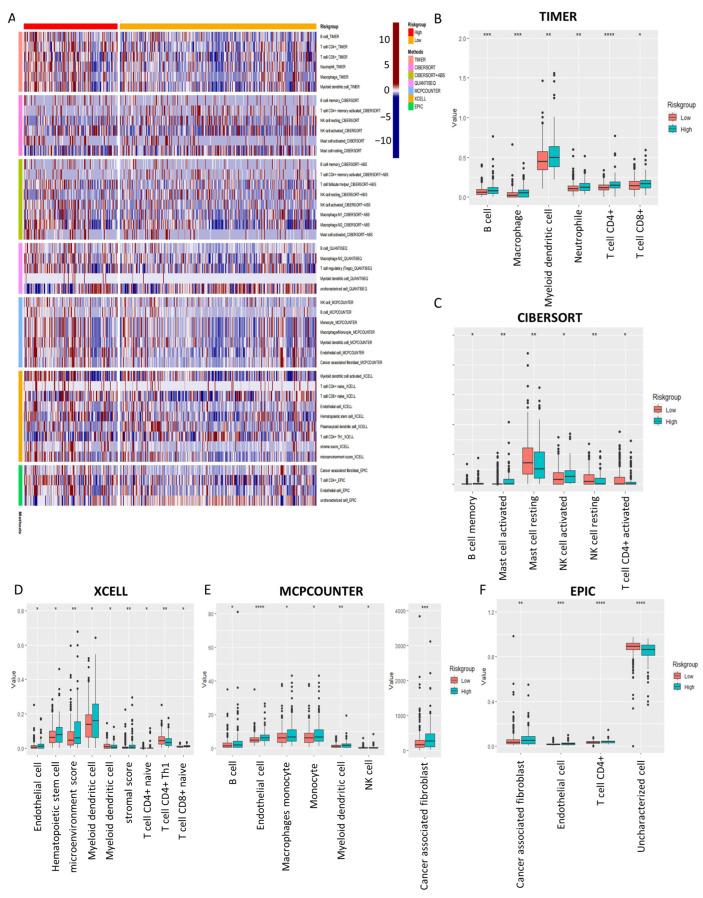
Analysis of immune cell infiltration landscape in COAD patients. (**A**) The heat map depicting an analysis of the immune cell infiltration differences between the groups at low and high risk via seven algorithms. The boxplots for the analysis of the difference for immune cell infiltration between low- and high-risk patients by TIMER (**B**), CIBERSORT (**C**), XCELL (**D**), MCPCOUNTER (**E**), and EPIC (**F**). *, *p* < 0.05; **, *p* < 0.01; ***, *p* < 0.001; ****, *p* < 0.0001.

**Figure 9 ijms-24-12915-f009:**
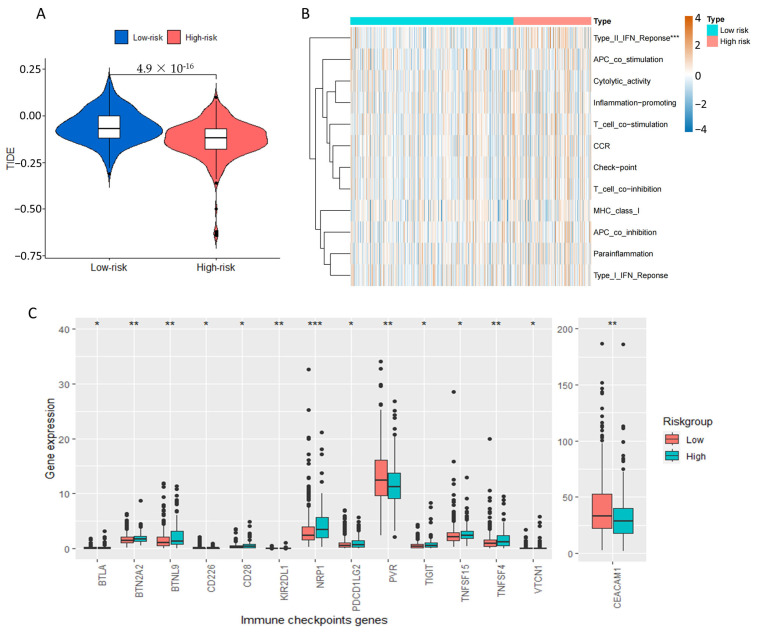
Analysis of tumor immune dysfunction and exclusion (TIDE), immune-related functions and immune checkpoints genes in COAD patients. (**A**) The violin plot of TIDE analysis for low- and high-risk group. (**B**) The heat map of the immune-related functions for low- and high-risk groups. (**C**) The boxplot of the expression levels of immune checkpoints genes for low- and high-risk groups. *, *p* < 0.05; **, *p* < 0.01; ***, *p* < 0.001.

**Figure 10 ijms-24-12915-f010:**
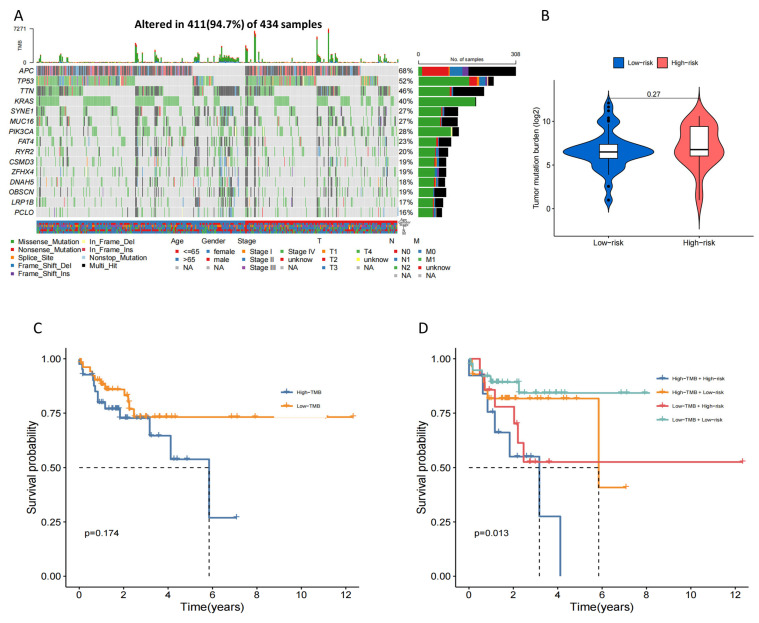
Tumor Mutational Burden (TMB) of the high- and low-risk groups. (**A**) The waterfall plot showed the TMB of top 15 genes in the combined set. (**B**) Analysis of the difference for TMB between low- and high-risk patients. (**C**) The Kaplan–Meier (K-M) survival analysis of COAD patients between high- and low-TMB groups. (**D**) The K–M survival analysis of COAD patients regarding TMB combined with risk score.

**Figure 11 ijms-24-12915-f011:**
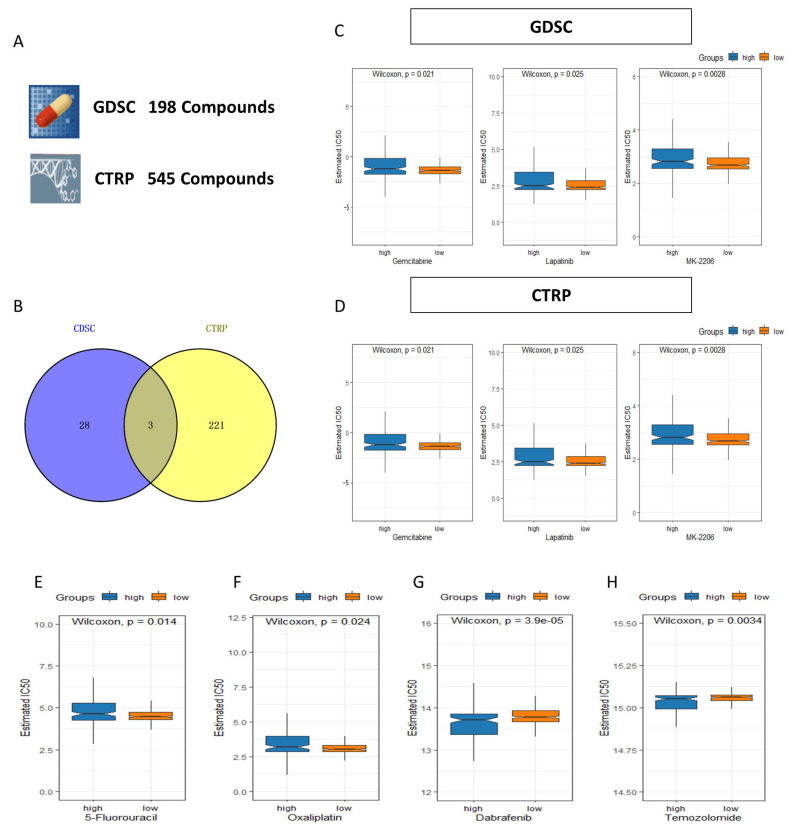
The drug sensitive prediction in COAD patients in low- and high-groups. (**A**) A total of 743 compounds from Genomics of Drug Sensitivity in Cancer (GDSC) and the Cancer Therapeutics Response Portal (CTRP) database were screened to investigate promising drugs for clinical treatment. (**B**) Venn diagram showing the three candidate compounds between GDSC and CTRP. Lapatinib, Gemcitabine, and MK-2206 all displayed a higher half-maximal inhibitory concentration (IC50) in the high-risk group both in GDSC (**C**) and CTRP (**D**). (**E**,**F**) 5-fluorouracil and oxaliplatin are the two compounds predicted with significantly lower IC50 in the low-risk group. (**G**,**H**) Dabrafenib and temozolomide are the two compounds predicted with significantly lower IC50 in the high-risk group.

**Figure 12 ijms-24-12915-f012:**
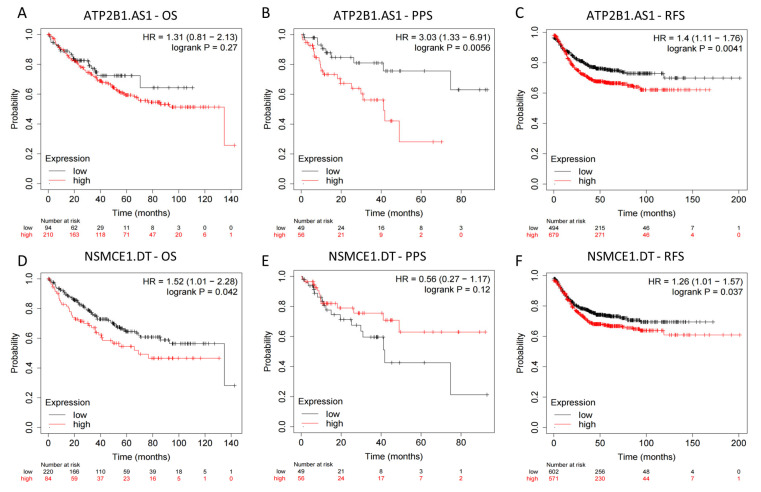
External datasets’ validation of DRLs as possible biomarkers. Overall survival (OS), progression-free survival (PPS), and recurrence-free survival (RFS) analysis of ATP2B1.AS1 (**A**–**C**) and NSMCE1.DT (**D**–**F**) from the Kaplan–Meier plotter database.

**Figure 13 ijms-24-12915-f013:**
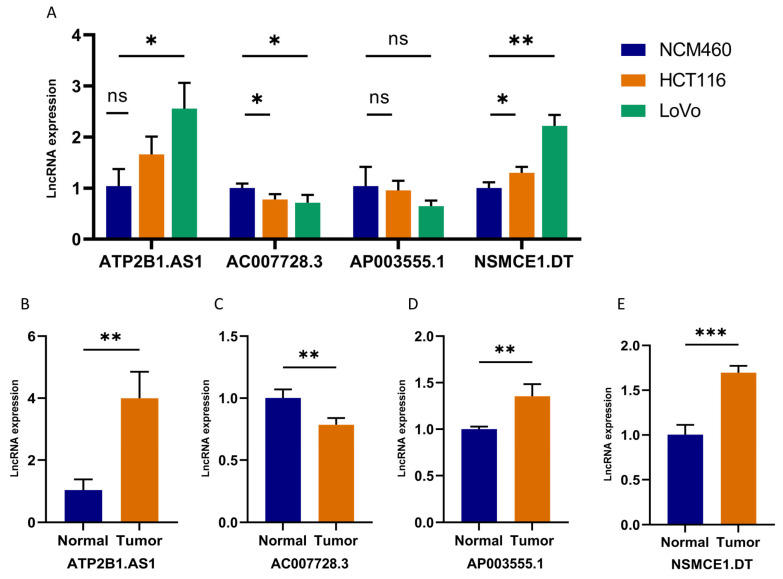
Validation of 4-DRLs expression in cell lines and tissues. (**A**) Comparison of the expression levels of ATP2B1.AS1, AC007728.3, AP003555.1, and NSMCE1.DT between NCM460 cells, HCT116 cells, and LoVo cells. (**B**–**E**) Expression analysis of ATP2B1.AS1, AC007728.3, AP003555.1, and NSMCE1.DT in tumor tissue samples and normal samples. *, *p* < 0.05; **, *p* < 0.01; ***, *p* < 0.001; ns, no significance.

**Table 1 ijms-24-12915-t001:** The clinical characteristics of colon cancer patients in training and testing set.

Characteristics	Total (*n* = 434)	Training Set (*n* = 217)	Testing Set (*n* = 217)	*p* Value
Age				
≤65	183 (42.17%)	98 (45.16%)	85 (39.17%)	0.2434
>65	251 (57.83%)	119 (54.84%)	132 (60.83%)	
Gender				
Male	233 (53.69%)	118 (54.38%)	115 (53%)	0.8473
Female	201 (46.31%)	99 (45.62%)	102 (47%)	
Stage				
Stage I	73 (16.82%)	39 (17.97%)	34 (15.67%)	0.1961
Stage II	166 (38.25%)	77 (35.48%)	89 (41.01%)	
Stage III	123 (28.34%)	68 (31.34%)	55 (25.35%)	
Stage IV	61 (14.06%)	28 (12.9%)	33 (15.21%)	
Unknow	11 (2.53%)	5 (2.3%)	6 (2.76%)	
T Stage				
T1	11 (2.53%)	6 (2.76%)	5 (2.3%)	0.6901
T2	75 (17.28%)	38 (17.51%)	37 (17.05%)	
T3	298 (68.66%)	152 (70.05%)	146 (67.28%)	
T4	50 (11.52%)	21 (9.68%)	29 (13.36%)	
N Stage				
N0	255 (58.76%)	127 (58.53%)	128 (58.99%)	0.8436
N1	102 (23.5%)	53 (24.42%)	49 (22.58%)	
N2	77 (17.74%)	37 (17.05%)	40 (18.43%)	
M Stage				
M0	321 (73.96%)	160 (73.73%)	161 (74.19%)	0.1552
M1	61 (14.06%)	28 (12.90%)	33 (15.21%)	
Unknow	52 (11.98%)	29 (13.36%)	23 (10.6%)	

## Data Availability

The original contributions presented in the study are included in the article/Appendix A. Further inquiries can be directed to the corresponding authors.

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
