# Peer review of "Construction and Validation of a Reliable Disulfidptosis-Related LncRNAs Signature of the Subtype, Prognostic, and Immune Landscape in Colon Cancer"

_ijms, 2023, doi:10.3390/ijms241612915_

Round 1

Reviewer 1 Report

The manuscript entitled Construction and Validation of a Reliable Disulfidptosis-related LncRNAs Signature of the Subtype, Prognostic and Immune Landscape in Colon Cancer”. The authors did extensive work, but a few questions arise from my side.

1.     What is disulfidoptosis, and how does it differ from other types of cell death in its response to thiol oxidation reagents like diamide?

2.     How does increased SLC7A11 expression in disulfidoptosis impact ferroptosis, especially in the absence of glucose?

3.     How does disulfidoptosis contribute to the anti-tumor immune response, and what potential does it hold for novel avenues in tumor treatment?

4.     How do dysregulated lncRNAs contribute to various aspects of colorectal carcinogenesis, including cell proliferation, apoptosis resistance, epithelial-mesenchymal transition (EMT), and immune evasion?

5.     In what ways can dysregulated lncRNAs serve as potential biomarkers for early detection, risk stratification, and treatment response prediction in colon cancer patients?

6.     How might targeting specific dysregulated lncRNAs offer a novel therapeutic strategy for combating colon cancer and overcoming drug resistance?

7.     What are the current challenges and opportunities in studying disulfidoptosis and lncRNAs in the context of colon cancer research?

8.     How many COAD patients were initially included in the study, and how were they divided into different groups?

9.     How many DRLs were identified based on Pearson analysis, and what were the criteria for selecting them?

10.  In the methodology section few updated references required to see (PMID: 35465161)

11.  What is the relation between DRGs and DRLs, as shown in Figure 2A?

12.  What method was used to explore the molecular subtypes of COAD based on DRLs, and how many clusters were identified?

13.  How many samples were included in each gene cluster, and what were the differences between these clusters as seen in the resulting heatmap (Figure 2B, C)?

14.  What were the clinical prognostic outcomes observed among the different subgroups based on gene clusters, as revealed by Kaplan-Meier survival analysis?

15.  Specifically, how did the patients in COAD cluster 3 differ from those in clusters 1 and 2 in terms of overall survival rates?

16.  Were there any significant variations in clinical traits between the training set and validation set of COAD patients, as presented in Table 1?

17.  How were the 434 COAD patients chosen from the TCGA database, and what criteria were used to ensure comprehensive clinical data for the study?

18.  Correct the Supplementary Materials text in black colour instead red.

19.  Which database was used to acquire somatic mutation data for the investigation of cancer-associated gene mutations?

20.  Did the total tumor mutational burden (TMB) significantly differ between the high-risk and low-risk groups?

21.  In terms of patient prognosis, how did the high TMB and high-risk group compare to other groups, and which group demonstrated the longest survival time?

22.  What packages or methods were used to analyze the chemotherapeutic drugs and search for potentially effective drugs?

23.  How many compounds were predicted for sensitivity across all patients, and what method was used to compare differences in drug sensitivity between the two risk groups?

24.  How many drugs from the CTRP (Cancer Therapeutics Response Portal) and GDSC (Genomics of Drug Sensitivity in Cancer) databases were included in the analysis?

25.  Were any specific drugs identified as showing significant differences in sensitivity between the high-risk and low-risk groups?

Good Luck!

Minor modifications are required. 

Reviewer 2 Report

Major concerns:

1.     Colon location and microsatellite instability are known factors that have varying levels of tumor infiltration, prognosis and are often diagnosed at differing stages. Were any of these considered when interrogating the structure of your initial clustering analysis? If not, why? Are your DLRs significantly associated with MSI/MSS? Are your results simply capturing genes that are specific to one of these tumor subgroups?

2.     DRGs were obtained from one previously published study. This seems like a limitation. Are these “real”, are these the only DRGs?

3.     What covariates did you include in your original COX regression analysis?

4.     The subgroup analysis (Figure 6) is interesting. However, without information on the number of samples within each subgroup, the data is hard to interpret. Please add n number, i.e. in your legend for each subplot to define the number of samples in SP high and SP low.

5.     Why was an absolute log2 FC change of 0.7 considered? This seems very arbitrary. What covariates were adjusted for in this regression analysis? If none, please explain why this would be appropriate.

6.     Line 323-324: you mention that AP0035555.1 is “improbable to be substantial”. This can be directly tested by removing the DLR and repeating some of your analysis. If it does not have an effect, then this is a fair comment to make. Otherwise, it seems difficult to justify.

7.     The data from potential compounds that may be effective against samples with high DLR expression would be more believable if an analysis of at least one of these compounds was performed in CRC cell lines and found to alter DLR expression.

8.     Please note that you can use RNA-seq data from CCLE to identify cell lines with (and without) your DLR signature for selection. With this in mind, why were the current cell lines considered appropriate for testing?

Minor Concerns:

1.     “In terms of occurrence, it is the third most prevalent cancer globally and holds the second position in terms of cancer-caused deaths”. Please rephrase. These statistics are for colorectal cancer, not colon cancer only. A recent 2023 study of cancer statistic states that CRC is the third leading cause of cancer related death:

Siegel, R.L.; Miller, K.D.; Wagle, N.S.; Jemal, A. Cancer statistics, 2023. CA: A Cancer Journal for Clinicians 2023, 73, 17-48, doi:https://doi.org/10.3322/caac.21763.

2.     Why was an analysis of age stratified at age 65?

3.     Line 249-251: “which is likely to explain why this group has poorer prognosis”. Please rephrase this. Without mechanistic interrogation of this process in your subgroup analysis, the writing is too strong a statement to make given your results.

4.     Immune checkpoint inhibition is also affected by MSI/MSS. Please consider accounting for tumor subgroup in your analysis of these genes.

Minor language checks:

1.     Line 86: “still empty”. Please rephrase.

2.     Line 88: “DRLs”. It is common practice to repeat abbreviations used in the abstract in the main text (as you have done for RCD).

Reviewer 3 Report

The study involves the identification and validation of a signature composed of disulfidptosis-related long non-coding RNAs (LncRNAs) to classify colon cancer subtypes, predict patient prognosis, and analyze the immune landscape within the tumor microenvironment. The researchers employ mainly bioinformatics techniques to analyze genomic data and develop a robust LncRNA signature that can potentially serve as a valuable tool for improving subtype classification, prognostic prediction, and guiding immunotherapeutic strategies in colon cancer patients. Overall, they identified a 4-DRLs-based signature, which seems to serve for forecasting the prognosis, immune landscape, and therapeutic response in colon cancer patients, thereby facilitating optimal clinical decision-making. Importantly, they use experimental verification of the signature's clinical relevance for further validation and application in the clinical setting.

Strengths of the Study:

1) The authors use sufficient and relevant data for the research question.

2) Their research question is clear and significant for the field, and the methodology followed is appropriate, well-described, and capable of answering the research question.

3) The statistical methods are suitable for the data and accurately applied.

4) The study is original and the findings contribute to the existing literature.

Minor comments:

1) Please comment on any potential biases or confounding variables that could influence the results.

2) Check if the results are accurately interpreted and supported by the data.

3) The authors are encouraged to discuss the implications of their findings and suggest future research directions.

Round 2

Reviewer 2 Report

Thank you. I have no more comments.